# Densely Packed Tethered Polymer Nanoislands: A Simulation Study

**DOI:** 10.3390/polym13152570

**Published:** 2021-08-01

**Authors:** Nicolas Chen, Oleg Davydovich, Caitlyn McConnell, Alexander Sidorenko, Preston B. Moore

**Affiliations:** 1Department of Chemistry & Biochemistry, University of the Sciences in Philadelphia, Philadelphia, PA 19104, USA; nchen@mail.usciences.edu (N.C.); cmcconnell@mail.usciences.edu (C.M.); a.sidorenko@usciences.edu (A.S.); 22720 Beckman Institute, 405 North Mathews Avenue, Urbana, IL 61801, USA; oleg2@illinois.edu

**Keywords:** polymers, surfaces, simulation

## Abstract

COordinated Responsive Arrays of Surface-Linked polymer islands (CORALS) allow for the creation of molecular surfaces with novel and switchable properties. Critical components of CORALs are the uniformly distributed islands of densely grafted polymer chains (nanoislands) separated by regions of bare surface. The grafting footprint and separation distances of nanoislands are comparable to that of the constituent polymer chains themselves. Herein, we characterize the structural features of the nanoislands and semiflexible polymers within to better understand this critical constituent of CORALs. We observe different characteristics of grafted semiflexible polymers depending on the polymer island’s size and distance from the center of the island. Specifically, the characteristics of the chains at the island periphery are similar to isolated tethered polymer chains (full flexible chains), while chains in the center of the island experience the neighbor effect such as chains in the classic polymer brush. Chains close to the edge of the islands exhibit unique structural features between these two regimes. These results can be used in the rational design of CORALs with specific interfacial characteristics and predictable responses to external stimuli. It is hoped that this the discussion of the different morphologies of the polymers as a function of distance from the edge of the polymer will find applications in a wide variety of systems.

## 1. Introduction

COordinated Responsive Arrays of Surface-Linked Polymer Islands (CORALs) are islands of densely packed polymers with lateral sizes and separation distances comparable to the polymer’s chain length [1]. They differ from classic homogeneous polymer brushes (PBs) with respect to their chain configuration and response to external stimuli. CORALs represent a unique and distinct organization of polymer chains tethered to the substrate. PBs are currently used in a myriad of research fields and various practical applications [2]. Applications of PB include dynamic control of adhesion and surface tension, polymer coatings, composite materials, colloids, joints, and micro- or nanomechanical devices, mechanical properties of nanocomposite materials, and stability of colloids [3,4,5,6,7,8,9,10,11,12,13,14]. CORALs are a promising next step to be used in these and other applications, because CORALs offer the promise of greater control and flexibility, and specifically with their ability to respond to the environment and external stimuli.

The characteristics of PBs, which include semiflexible polymers due to confinement, such as their grafting density, density profile, and conformation of grafted polymer chains have been the focus of intensive research over several recent decades [15,16,17,18,19]. CORALs are sometimes, and we believe incorrectly, considered as islands of PBs. While CORALs are composed of islands of densely packed polymers similar to PBs, CORALs have unique and distinct structural properties. For example, the island edge, which creates a responsive soft interface between the substrate surface, grafting polymer, and solvent, is a distinctive functional feature of CORALs not present in PBs. A large fraction of the polymers within CORALs are at this boundary due to the nanoscale of the islands. Therefore, CORALs are structurally and dynamically distinct from polymer chains in the island. Further, the polymers in PBs all exhibit the same functional properties, while in CORALs different polymers within the same nanoscale island behave differently.

Due to their dynamic structure, CORALs are able to adopt distinct configurations. For the CORALs presented, two distinct conformations are observed: A flat configuration which occupies the entire surface of the substrate (relaxed state, RS), and a compact state (CS), where the polymers within an island compact to exposes a substantial portion of the substrate (Figure 1 and Figure 2). Thus CORALs demonstrate dynamic, reversible morphological rearrangement. Materials, such as CORALs, which reversibly change their morphologies and surface properties in response to external stimuli are sometimes referred to as “smart materials” [20,21]. This unique property is a critical distinguishing feature of CORALs and could provide results unattainable with typical PBs.

The investigation herein continues the exploration of tethered polymers’ behavior, with a focus on a single component of CORALS: that of isolated tethered polymer islands. To assess the polymer chains that constitute these islands, we consider fundamental parameters that have previously been used to describe grafted polymer chain configurations: grafting density (σ), reduced tethered density (Σ), polymer height (*h**), polymer thickness (*h*), radius of gyration, ($R_{g}$) and its components or ratios of its components such as ($R_{g_{Z/XY}}^{2}$). Further, as the islands are of the same size as the grafted polymers, we consider different inland sizes, and how the polymer parameters vary within these different inland sizes. Foreshadowing the results, we find different characteristics of semiflexible polymers: at the edge of the island we find flexible polymers, a few grafting sites from the edge we find the polymers which exhibit confinement and semiflexible characteristics, and in the middle of the island we find polymers with similar characteristics to polymer brushes.

There is a long history of simulating polymers and PBs [22,23,24]. A review that captures the utility and benefits of computer simulations to understand the dense polymer systems states Molecular Dynamics (MD) simulations are critical “to test theory and interpret experiments” [25]. A major advantage of MD simulations is their ability to provide insights into structure and dynamics at the molecular level within materials by providing access to the positions and velocities of individual particles not easily obtainable through experiments. As a result, the relevant microscopic mechanisms of particular phenomena can be clarified and/or predicted in a way that is difficult or not possible through either theory or experimentation [26,27].

MD simulations of dense polymer systems can be split into three categories: (1) atomistic, where the interaction site represents an atom or a heavy atom with hydrogens (e.g., CH_2_), (2) coarse-grain (CG) and mesoscopic, where the interaction site represents a fragment of a polymer chain or a collection of solvent molecules (e.g., four water molecules, or three CH_2_ moieties) [28], and (3) continuum methods [29] such as those employed in fluid dynamics. We adopted a mesoscopic approach that uses a simplified forcefield, yet still captures the essence of the processes we sought to investigate. Although this approach obscures some atomistic details by focusing on moieties and segments, we are capable of investigating phenomena such as microphase separation and dynamic rearrangement within CORALs. To our knowledge there have been no other simulations of CORALs; however, there have been simulations of nanoscale PBs [30] and their responses to solvent [31]. Léonforte and Müller [31] simulated PolyAcrylic Acid (PAA)/PolyStyrene (PS) brushes using CG modeling under different solvent conditions. Different patterns and morphologies were found depending on the solvent and underlining grafting topology and, while similar topologies are seen in CORALs, the polymer chains within tethered islands are unique to CORALs and have not been characterized to our knowledge.

## 2. Materials and Methods

For these systems, we pursued the use of a mesoscale model that is simple yet captures the essential physics. It is imperative to accurately depict the physical behavior of polymer chains, without trying to replicate a specific polymer. In the model, the use of beads to represent several monomeric units was chosen with the monomeric units or polymer unspecified. Each bead represents one Kuhn length segment of the polymer and subsequently allows for the formation of self-avoiding freely jointed chains. Thus these simulations can be used for multiple different polymers, for example polystyrene has a Kuhn length of 6 monomer units [32] per segment length, thus σ≈ 0.55 nm in SI units, and for PolyLactic Acid (PLA) the Kuhn length is 8–10 units, which corresponds to σ≈ 17 nm [33] in SI units. Beyond the intramolecular bead spring model, we model the intermolecular interactions as a Lennard Jones (*LJ*) 12–6 potential (see Equation (Equation 1)).
(1)$$V\left(r\right)=4\epsilon ({\left(\sigma_{LJ}/r\right)}^{12}-{\left(\sigma_{LJ}/r\right)}^{6})$$

The *LJ* parameter $\sigma_{LJ}$ between the solvent moities is used for our length unit and ϵ for our energy unit. In the spring model a harmonic potential between sites was used where $V\left(r\right)=k_{bond}{\left(r-r_{0}\right)}^{2},k_{bond}=10\epsilon$, and $r_{0}=2^{\frac{1}{6}}\left(2\sigma_{LJ}\right)\approx 2.245\sigma_{LJ}$, such that the bonding and *LJ* potential have the same distance at the potential minimum. Other intramolecular interactions may be used to model increasingly detailed molecular structures using more complex potentials, an example being FENE bonding, angles, and torsions. However, the aforementioned models all provide similar structure and scalings (e.g., $R_{g}$ scaling in bulk) and are typically fine-tuned for specific polymers and not used here. The complete description of all the intermolecular interactions in our system are given in Table 1.

The liquid density is chosen such that the pure solvent is well within the liquid phase part of the phase diagram, more specifically bulk solvent density and temperature are $\rho^{*}=0.6$ and $T^{*}=1.2$. The surface, or “wall”, is modeled by using stationary sites in two planes that interact with the polymer and solvent but do not interact with other wall sites. The fixed wall particles are spaced every $0.25\sigma_{LJ}$ within the plane, which were separated by $8\sigma_{LJ}$, therefore preventing solvent or polymer sites from passing through the wall. The polymer chains were tethered in a square pattern at each of the 16 wall sites for spacing (*D*) of $4\sigma_{LJ}$, (Figure 3).

We used LAMMPS [34] to perform the simulations on AMD or INTEL cores. The initial structure was equilibrated using the NPT ensemble (temperature: $T^{*}=1.2$, pressure: $P^{*}=0.25$ for the solvent particles, and providing a bulk density of $\rho^{*}=0.6$). When the wall was incorporated we allow only the *z* coordinate to fluctuate which maintains the appropriate density of the solvent phase. We identified each equilibrated system when the values of the radius of gyration of each polymer, density profiles, elements of the pressure tensors, and the cell dimensions, all fluctuate around a constant average value. Once equilibrated the system was simulated using the NVT ensemble for at least 5 million steps was collected for analysis. Furthermore, to ensure equilibrium within the production runs, we use block averaging, where we averaged the first 1/2 of the simulations, the second 1/2 of the simulations as well as the entire simulations. All these averages and their standard deviation were consistent.

We validated and characterized our polymer model by analyzing simulations of systems with a single chain of different lengths (e.g., number of polymer beads (*N*)) in bulk solvent (i.e., no wall). This allowed us to firmly establish the scaling of $R_{g}$ as a function of polymer length or number of polymer sites *N*. Within a theta solvent (i.e., where the polymer acts as a freely jointed chain) $R_{g}$ increases proportional to $N^{1/2}$ ($R_{g}\propto N^{1/2}$). We obtain this scaling with a polymer–solvent interaction of 0.7ϵ. We also characterized a weak solvent, with ϵ=0.5, where $R_{g}$ scales as $\approx N^{1/3}$, and a good solvent, ϵ=1, where $R_{g}$ scales as $\approx N^{2/3}$ From Figure 4) we confirm that the scaling of $R_{g}$ goes as $\propto {\left(N/2\right)}^{1/2}$ as expected for a freely jointed chain.

### Model Validation

Once the interaction for a theta solvent was validated, we compared a single polymer in solution with one that was linked to a wall. We find, as expected, that, while the $R_{g}$ of the link polymer is similar to the single polymer in bulk, it can be perturbed with different wall/solvent and wall/polymer interactions. For simplicity, we chose a wall with both wall/solvent and wall/polymer interactions of 0.1ϵ. This weak interaction ensures that the wall interactions are neither too attractive nor too repulsive. In future studies different interaction parameters between the wall and the solvent/polymer as well as good and weak solvents which will be explored. Varying these different interactions affects island morphology and the polymers’ response to external stimuli such as changing solvents. The current study is critical to further studies as it will serve as a reference point for which other simulations can be compared.

We chose interaction parameters and an isolated polymer island model system to mimic the critical component of the tethered polymer island of an experimental CORAL system we have reported on earlier [1]. We choose polymers length (*N*) of 40 sites which was the smallest *N* which had undetectable finite size effects in bulk scaling simulations (Figure 4) and a distance between grafting points (*D*) of $4\sigma_{LJ}$, which corresponds to a brush regime. Specifically $R_{g}^{2}$ is $94\sigma_{LJ}^{2}$ for the isolated chain tethered to the wall, well within the brush regime. With appropriate scaling, these conditions corresponds well to the experimental AFM data of tethered islands consisting of diblock copolymer (35.9 kDa polystyrene tethered via a minor 4.4 kDa poly-4-vinylpyridine block) with a height to radius ratio of 2/3 [1].

With our model and interactions chosen, we endeavored to simulate an isolated island of grafted chains, to understand this critical component of the CORALs. Our grafted islands have two limits in terms of the number of polymer chains within the island: that of the single isolated grafted chain as the “smallest” island and an “infinite island” which corresponds to the standard PB. Between these two limits are islands with the same grafting density but different molecular geometry when compared to a PB. For this study we consider islands of varying sizes, specifically islands with 1, 16, 32, 80 and 524 grafted polymers (Figure 3), along with a periodic system representing the full brush for comparison.

## 3. Results and Discussion

We simulated isolated islands of different sizes to characterize the polymer structure as a function of island size and the results are summarized in Table 2. The single tethered chain, the smallest island possible, is spherical or slightly oblate (pancake) in shape. This is due to the interaction with the wall and the solvent–wall structure, such as the layering of solvent at the wall thus flattening the structure relative to the bulk spherical shape. As expected for a theta solvent, the polymer expands in all directions except for into the wall, i.e., a self-avoiding freely jointed chain. Tethered chains within an island (more than a single chain) experience crowding by neighboring chains which pushes the polymer structure toward a prolate shape and into a “brush” like configuration, e.g., semiflexible. However, if the island is small then the polymer crowding is never enough to fully extend the system into the brush regime, because the polymers are able to extend beyond the island where there are no tethered polymers and no steric repulsion. The transition from isolated chains to semiflexible occurs when the island size is increased to a size that polymers with the middle of the island no longer are influenced by the edge of the island and are similar to a PB. The transition is monitored by different height characteristics as well as components of the radius of gyration and their different components.

We characterized the polymer and island structure through a series of parameters. The maximum height ($h^{*}$) is polymer site obtained over the wall, which would be similar to what the AFM data would obtain. The thickness (*h*) characterizes when the average density approaches zero, i.e., when you would not expect to see any polymer. Tethered density Σ which characterizes the crowding of each polymer was also used. The radius of gyration $R_{g}$ which gives the extent of the polymer, and the ratio of the $R_{g}^{2}$ components *Z/XY* which give a shape to the polymers were also used. We discuss each of these parameters in turn in depth below.

### 3.1. Max Height (h*)

We calculate the height profile of each island to determine their structures. For this calculation we gridded the surface and plotted the highest polymer bead within each grid, which was then averaged over all of the configurations. These data are similar to what one would expect from AFM data, i.e., when the AFM tip would interact with the island from above. Figure 5 shows the height profile for islands of different sizes. As expected, the shape of the overall island is roughly commensurate with the tethering sites with the maximum height located in the center of the island. The polymers near the edge extend outward beyond the edge of the island allowing for less crowding and a greater “pancake” shape of the entire island. The polymers in the center of the island do stretch more in the *z*-direction because of neighboring polymer interactions (crowding) and the lateral polymer heterogeneity in the *xy*-direction (island edge) is observed. The polymers at the edge expand in the *xy*-plane, parallel to the wall, resembling similar behavior to a single tethered chain. We find this to be a common theme. The edge polymers act similar to a single tethered chain, while the center polymers are similar to polymers in a brush. We find polymers in an intermediate regime one to two graphting layers removed from the edge. Interestingly, the crossover to this regime requires multiple neighbor layers and is not fully realized until there are 10 layers of polymers between this region and the edge of the island.

### 3.2. Height Profiles (h)

While the max height profiles show the extent to which the polymer extends above the surface, it does not provide information on how the density within the island changes. To understand how the density changes we report the polymer density profile (Figure 6) as a function of distance from the wall and edge of the island. Experimentally, this density could be obtained with X-ray scattering data. We see density oscillations occurring close to the wall due to polymer packing as expected. It is noteworthy that PB densities are very different compared to the islands, due to the lateral homogeneity in the PB and the heterogeneity in the island systems.

Using the equations introduced by Cates et al. [18], we fit the monomer density profile, ϕ(z), as a function of distance (*z*) from the surface $\phi \left(z;r\right)=\left(B/w\right)\left(h^{2}-z^{2}\right)$, where $\left(B/w\right)*h^{2}$ is the density at the surface, and r is the region within the system that is being analyzed. This equation predicts (for a uniform grafting polymer such as a brush) a parabolic density profile which goes to zero at a thickness of *h*. For the *h* values reported in Table 2, we use the entire islands to obtain an average “height”. As seen in Table 2, these values closely follow the hmax calculated previously. Further, monomer density (Figure 6) resembles a parabola for larger islands, as predicted by Cates et al. [18]. However, significant deviations arise in smaller islands due to the density of monomers at the edge which are not the same as within a uniform grafted polymer which was assumed by Cates in deriving the equation.

To analyze the difference height profiles we divide each island into three regions: central, edge, and outside (see Figure 7). The central region is defined as having *xy* coordinates between the middle and 50% of the island radius, the edge region is the outer region where *xy* coordinates give a radius between 50% and 100% of the island and the outsize region is anything that is outside the island radius. For illustrative purposes we focus on the island with 80 tethered chains, as this illustrates the common themes we see in all the island sizes. Within the central region, the density profile closely matches that of the predicted polymer brush. However, at the edge and outside, the density maximum (Figure 7) is not at the wall surface, as would be expected, and indicates there is more density away from the surface then directly next to it. Using a parabolic function is not appropriate for these regions as the the assumption of a homogeneous grafting density is not valid (e.g., there is more polymers toward the islands center then away from the island center). Different functional forms would need to be employed to fit these density profile, and (Figure 7) is a good illustration of a need for a different theory of monomer density in these systems. This is the subject of future work, but is beyond the scope of the present manuscript. The reason for the maximum being away from the wall is because the solvent has no steric hindrance outside the edge, Thus, there is a higher solvent density near the wall and commensurate lower polymer monomer density near the wall. The monomer density then peaks before smoothly decaying to zero. The solvent density is inversely correlated with the monomer density. Again, these observation provide insight that the three regions (middle, close to the edge and edge), behaving differently.

### 3.3. Radius of Gyration (R_g_)

$R_{g}$ and $R_{g_{Z/XY}}$ are used to characterize the polymer shape as a function of position within the island. The chain shows brush characteristics when the ratio *Z/XY* is greater than one, otherwise the chain is in mushroom or pancake state. This ratio requires the decoupling of the components of $R_{g}$, unlike experiments, computational calculation access to such information straight forward. All island systems show brush characteristics at the center of the island with *Z/XY* greater than one, intermediate characteristics near the edge, and the chains show mushroom conformation at the edge similar to the isolated tethered chain, as shown in Figure 8. The $R_{g_{Z/XY}}$ parameter could be used to compare to a parameter called reduced tethered density (Σ) used in experiments, which we are going to discuss intensively in the following section.

It is noteworthy that CORALs islands have all tethered polymer regimes simultaneously, from the interiors of the islands polymers being semiflexible in the “brush” regime to the “isolated” tethered chain at the island’s edge.

### 3.4. Reduced Tethered Density (Σ)

One of the key characteristics of PBs are their grafting densities. Minko and Brittain [4,35] have proposed a universal parameter of reduced tethered density (Σ) to distinguish grafted polymer films of different densities: Σ is a reduced parameter which depends on the chains changing conformation under the same conditions (polymer nature, temperature, solvent, etc.) as an tethered isolated polymer chain. Specifically, $\Sigma =\pi \sigma R_{g}^{2}$, where $R_{g}$ is radius of gyration of an isolated tethered chain at specific experimental conditions and σ is the grafting density. We use σ as ∝(h) thickness [35] or for a uniform distribution of points becomes $1/D^{2}$, where *D* is the distance between grafting points. The “brush” regime of PB has been empirically established to occur at Σ>5 where the grafting density is significantly high such that polymers interact and stretch away from the surface (e.g., semiflexible polymers). Sparsely distributed Σ<1, wherein the polymer chains do not interact with one another. The sparsely distributed case has been explored in several relevant publications by Genzer [36], Santore [37,38], Russell [39], and Sommer [40]. However, critically for our system, we do not have uniformly distributed tethered chains, but tethered heterogeneity, e.g., the coverage of the surface is not the same everywhere. Instead of a uniform grafting density, we have both high local density and low (or zero) local grafting density, with the average grafting density somewhere in the middle. These high and zero grafting densities are by design. Furthermore, this heterogeneity is on the same scale (≈1/4) of the stretched polymer length. The tethering is not randomly distributed or even randomly distributed in different areas, but the high density islands create an array pattern on the surface, with zero density between the islands from the masking during synthesis. This special geometry and morphology make CORALs a unique material and distinct from other tethered systems such as polymer brushes.

Since the grafting is heterogeneous, we calculate a local $\Sigma_{L}$ for every individual polymer. We define the local reduce tethered density using $\Sigma_{L}=\pi R_{g}^{2}/D^{2}$, where *D* is the grafting density (e.g., $\sigma =4/D^{2}$). Since in our simulation we use a square pattern for polymer grafting, we add a correction factor to the grafting density (e.g., $\sigma =4/D^{2}$) so we can include the packing factor contributions to the calculation of $\Sigma_{L}$. Figure 9 shows the structure of the polymer at its relative location on the polymer island. The polymers generally show large $\Sigma_{L}$, or brush-like characteristics, due to the close grafting points. Consistent with our analysis of $R_{g_{Z/XY}}$, the PB characteristic increases as the polymers are closer to the center of the island as expected. In our study, as the island size increases, the PB characteristics are greater, e.g., $\Sigma_{16-polymer-island}<\Sigma_{32-polymer-island}<\Sigma_{80-polymer-island}<\Sigma_{524-polymer-island}$.

### 3.5. Island Density $\Sigma_{island}$

We measure the average island $\Sigma_{island}$ in two different ways. First we determined $\Sigma_{island}$ using the thickness (*h*) of the island (e.g., =σ∝h) and the results are shown in Table 2. Second, we used the distance between the grafting points ($\sigma =1/D^{2}$) to calculate the $\Sigma_{island}$ by averaging the $\Sigma_{L}$ in the island, as shown in Figure 9. These results are consistent. For example, with the 80-polymer island, method 1 reported 7.7, whereas in Figure 9 $\Sigma_{island}$ is in the range of 6–8 for the second method. As these methods are averaged over all polymers it is reasonable that they give similar results.

All systems we studied would be characterized as in the brush regime since $\Sigma_{island}>5$ for all cases. Although the island exhibits brush characteristics (e.g., $\Sigma_{island}$ > 5), we find that PB characteristics are observed within center of the island and decreasing toward its edge. At the edge of the island, the chain characteristics deviate substantially from typical PBs (see Figure 9). We find that the brush regime is obtained within one or two tethered points from the edge of the islands. Thus the polymers structure is influenced strongly by neighbors as expected. However, to fully reach the brush regime many neighboring shells are required. This can be seen that the islands with 16–80 polymers, while in the brush regime they never converge to the infinite case. The island with 524 polymers does converge to the brush regime as seen in (Figure 9). The change from 16, 32, 80 to 524 is not gradual; one would expect that the edge of the islands look the same; however, the confinement and steric hindrance is mitigated by the size of the island where the polymers extend beyond the edge. Thus the brush like character of the interior rises with size of the island until one reaches the infinite limit.

### 3.6. Semiflexible Polymers

In this Special Issue on semiflexible polymers, where “chains serve as a coarse-grained representation of macromolecules whenever random or self-avoiding walk statistics do not apply” are particular applicable for these isolated densely pack islands, and our simulations. We used a mesoscale representation of the polymers such that our results can be mapped onto any system with a similar geometry. The tethered polymers are influenced by the surface, and neighboring tethered polymers, which strongly perturb their shape. Within these systems we see polymers exhibit different perturbations depending on their position within the islands, with polymers on the edge of the island essentially unperturbed from a isolated single tethered chain, to polymers in the brush regime that are strongly perturb by neighboring polymers. We focused on the statics and simulation of these unique materials which have many different characteristics of semiflexible polymers. As our simulations use scaled interactions, the results themselves can be scaled to many different systems. It is hoped that this discussion of the different morphologies of the polymers as a function of distance from the edge of the polymer will find applications in a wide variety of systems.

## 4. Conclusions

CORALs are novel systems containing polymer chains tethered periodically onto a substrate as isolated nanoislands of densely packed polymers susceptible to environmental conditions. We characterized the structural features of the chains while in the relaxed state. Within these islands are differential regimes of tethered polymers exist. These polymers exhibit different semiflexible characteristics as a function of their distance from the edge of the island. Within the interior of the island are polymer chains with characteristics more consistent with a dense polymer brush. This behavior contrasts the peripheral chains along the island’s edge which behave similar to isolated chains. It should also be noted that between these opposing behaviors, an intermediate region exists and forms a zone of transition from the brush-like regime to those demonstrating traits of isolated chains.

This study involved the use of simulations with interaction parameters that can be scaled to experimentally realized CORAL systems consistent. The simulation parameters reveal insights into the interaction needed for rational design of materials with specific properties and characteristics. The system is made to allow for a great variety of modifications and to also accurately predict the different possible morphologies, properties, and responses. Obvious extensions beyond the current study include: interactions of the solvent and polymer with each other and the substrate, and the length of the polymer chains and their tethering density within the island. These parameters will be explored in future simulations. The results reported herein serve as a baseline against which to compare future work, with the ability to model specific materials with distinct, customizable properties and characteristics.

## Figures and Tables

**Figure 1 polymers-13-02570-f001:**
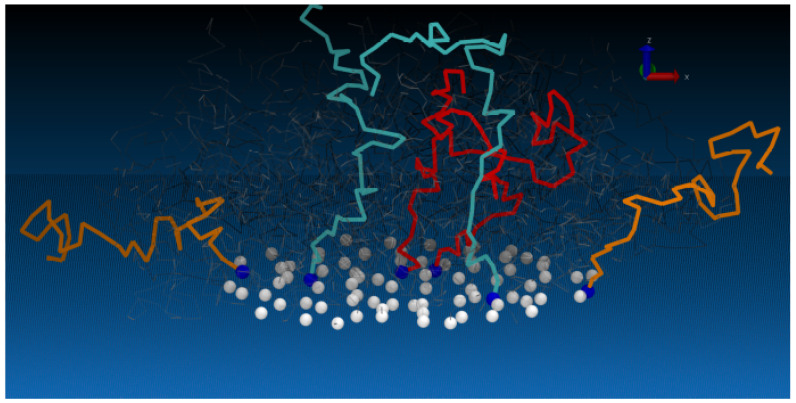
Abstract Figure: a snapshot from a simulation showing different characteristics of semiflexible polymers depending on the distance from the edge of the island. Polymers on the edge are more flexible and extended than the polymers within the interior, which behave as a polymers in the polymer brush regime. Polymers near the edge but not at the edge have characteristics between these extremes.

**Figure 2 polymers-13-02570-f002:**
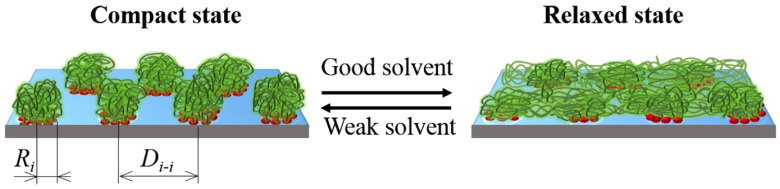
CORALs: an artistic representation, showing the coordinated morphological switching of surface-linked polymer islands (from CS to RS). These switches occur in response to “good” solvents and reverse in response to weak solvents. $R_{i}$ and $D_{i-I}$ are the island radius and inter-island distance, respectively.

**Figure 3 polymers-13-02570-f003:**
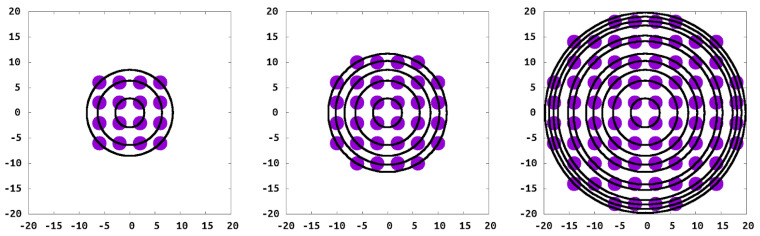
Grafting points of the 16, 32, and 80 polymer islands, grafting coordinates with the center or the island at the origin.

**Figure 4 polymers-13-02570-f004:**
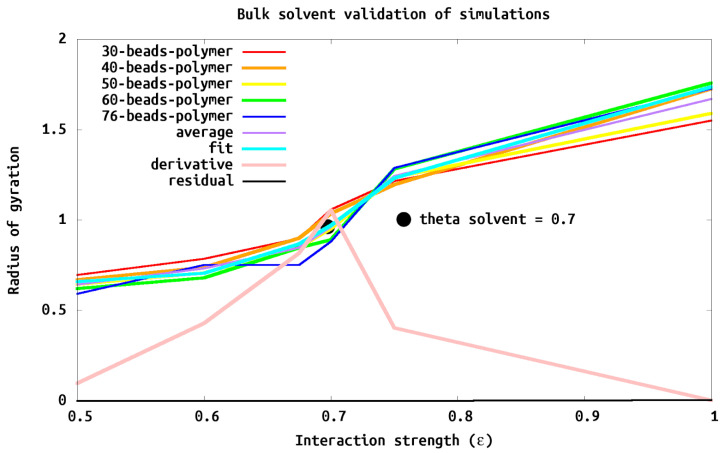
Bulk solvent validation of simulations to determine theta solvent. Different length polymers (*N* = 30, 40, 50, 60, 76) and interaction strengths (ϵ = 0.5, 0.6, 0.675, 0.7, 0.75, 1.0) were simulated in bulk solvent. Each $R_{g}$ was scaled by $\sqrt{\left(N/6\right)}$ such that a theta solvent should be 1 on the *y* axis. The average is the *R_g_* of all the simulations of each polymer at a particular ϵ. The derivative is calculated as the numerical derivative of the average, where the maximum would be the ϵ corresponding to a theta (θ) solvent. We determine (and see) that ϵ = 0.7 behaves as a theta solvent for this system regardless of Molecular Weight (e.g., length of polymer *N*).

**Figure 5 polymers-13-02570-f005:**
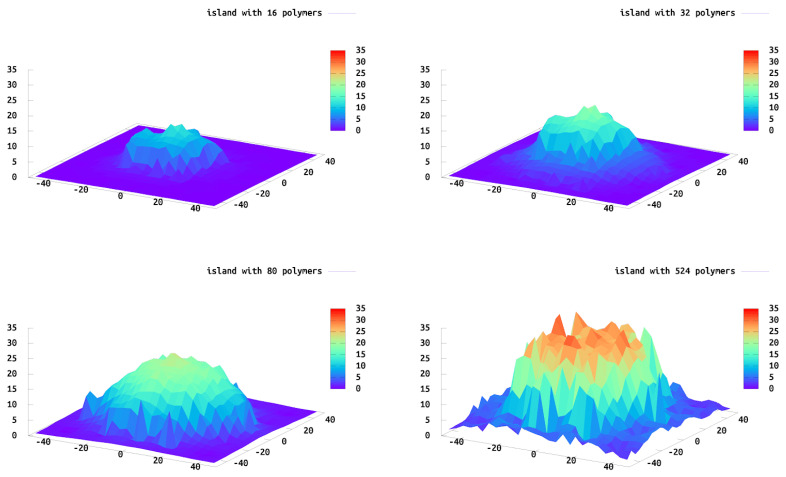
Isolated island height profiles of 16, 32, 80 and 524 tethered polymers, respectively. The highest polymer bead within each bin defines the height of the polymer, averaged over 5000 configurations.

**Figure 6 polymers-13-02570-f006:**
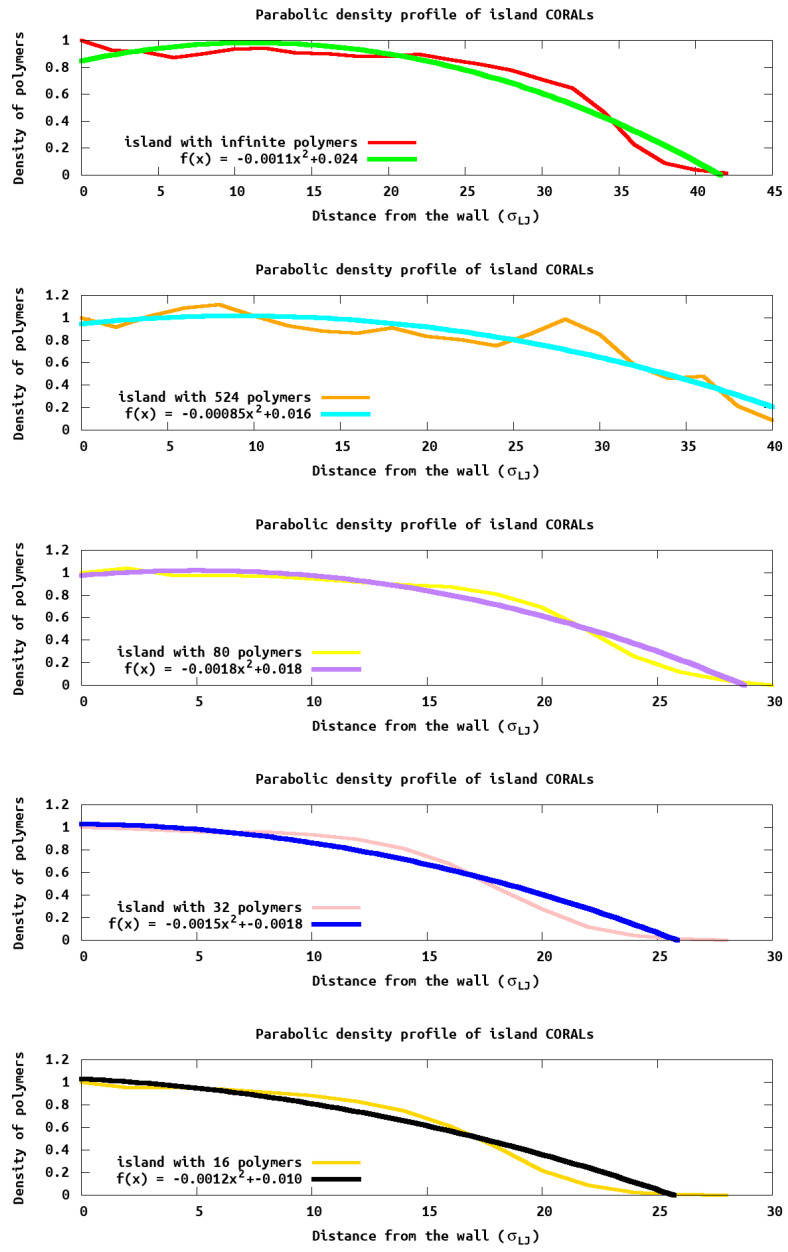
Parabolic density profile: Fitting the reduced density profile with a parabolic function provides a robust definition of the thickness of the island. Here the density of height of each island is plotted verse the height, e.g., distance away from the wall.

**Figure 7 polymers-13-02570-f007:**
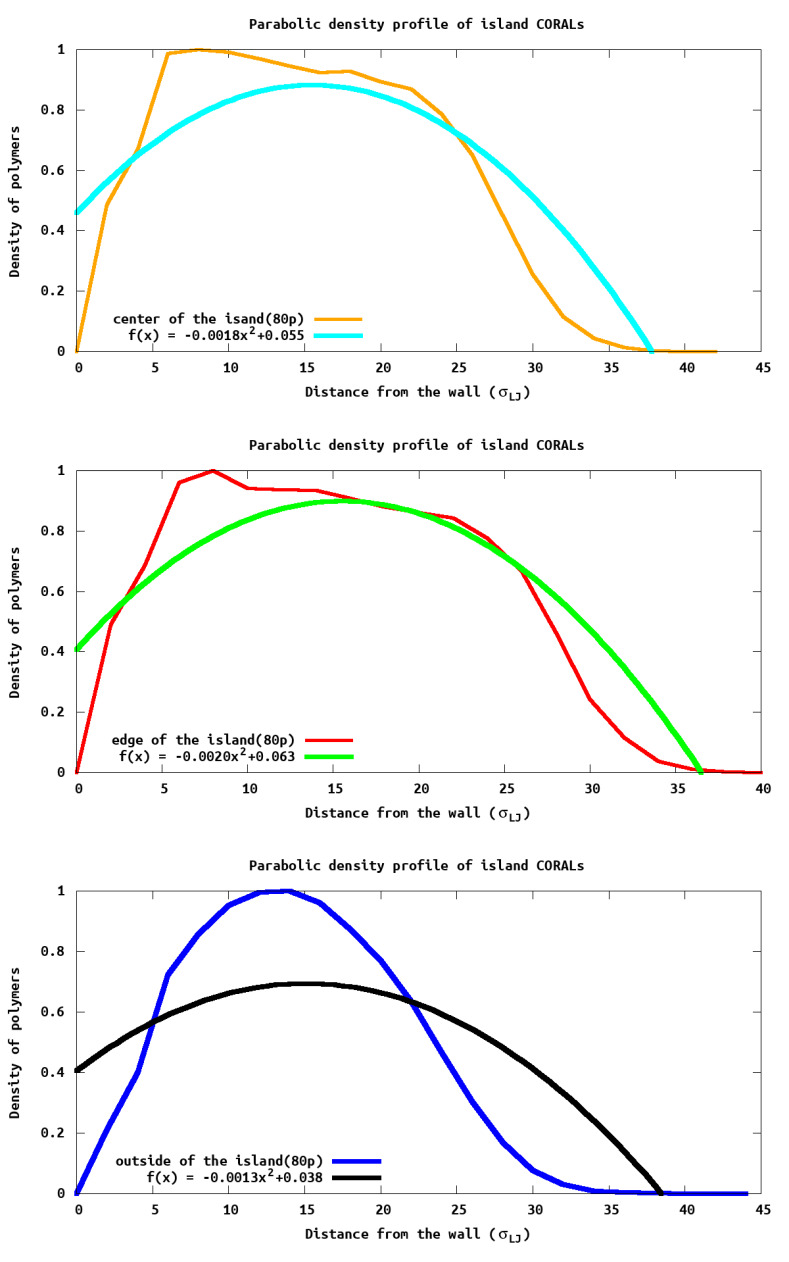
Parabolic density profile of different regions of the island. Fitting the reduced density profile with respect to different regions of the island clearly show the center, edge, and the outside of the island having different characteristics.

**Figure 8 polymers-13-02570-f008:**
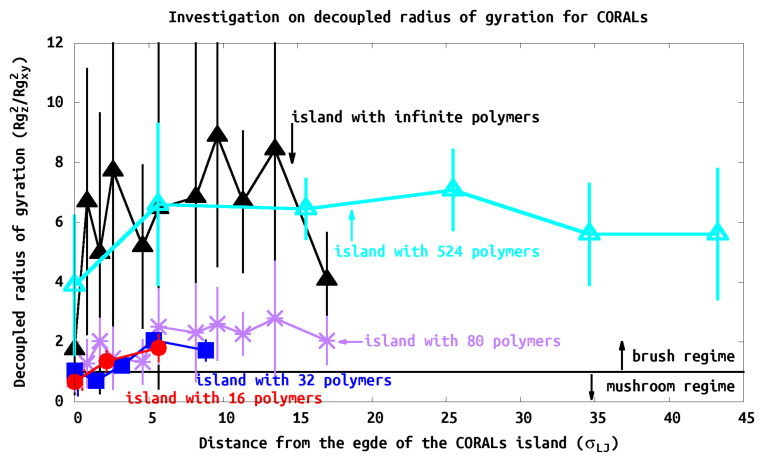
Ratio of the decoupled radius of gyration as a function of distance from the edge of the island. The ratio increases as the polymers approach the center of the island.

**Figure 9 polymers-13-02570-f009:**
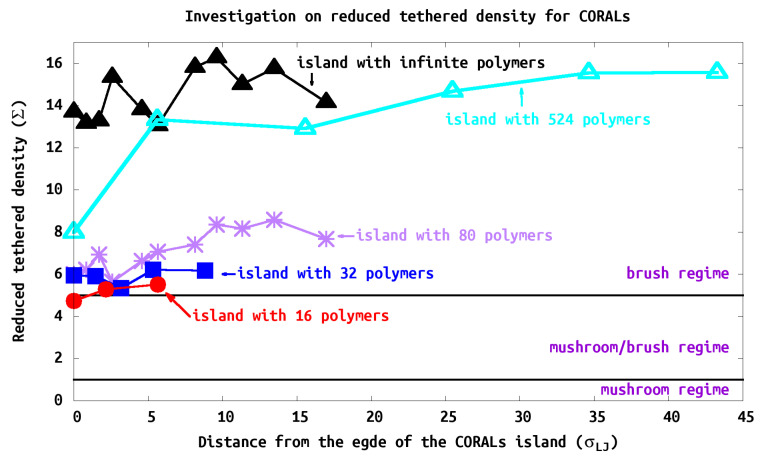
Tethered density profile of CORALs. All systems show strong PB characteristics, e.g., $\Sigma_{island}$ > 5. As the size of the island increases, the reduced tethered density increases. When approaching toward the center of the island, polymers exhibit stronger PB behavior.

**Table 1 polymers-13-02570-t001:** Interaction Parameters.

Interaction	Polymer		Solvent		Wall	
	$\sigma_{LJ}$	ϵ	$\sigma_{LJ}$	ϵ	$\sigma_{LJ}$	ϵ
Polymer	1	2	0.7	1.5	0.1	1
Solvent			1	1	0.1	1
Wall					NA	NA

**Table 2 polymers-13-02570-t002:** Systems simulated and summary of results (the number of sites per polymer was 40, length in $\sigma_{LJ}$ units).

Polymers	Island Radius	Particles	Max Height $h^{*}$	Thickness *h*	Tethered Density Σ	$<{R_{g}}^{2}>$	$<{R_{g}}_{Z/\mathrm{XY}}^{2}>$
1	NA	670,864	8.97	NA	NA	122 ± 87	0.02 ± 0.02
16	6	654,216	18.63	17.86	5.05	33 ± 5	1.5 ± 0.6
32	10	725,412	22.63	18.43	6.41	38 ± 6	1.5 ± 0.6
80	18	1,339,914	24.45	21.40	7.71	46 ± 16	1.9 ± 1.0
524	50	1,887,092	33.80	33.89	12.86	119 ± 49	5.8 ± 1.9
∞	NA	1,472,000	34.06	31.54	15.63	92 ± 34	6.8 ± 4.5

## Data Availability

The data presented in this study are available on request from the corresponding author.

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
