# Peer review of "Densely Packed Tethered Polymer Nanoislands: A Simulation Study"

_polymers, 2021, doi:10.3390/polym13152570_

Round 1
Reviewer 1 Report
In this article, the authors present a Molecular Dynamics simulation study of polymer islands (CORALs) of various sizes. A CORAL is an array of densely grafted polymers on a planar substrate with lateral size comparable to the length of a polymer chain. Their finite size makes them distinct entities from the polymer brushes. CORALs can have interesting applications, and their study is an important subject of research . This article is a significant contribution in the description of their conformational properties. I will recommend publication after the authors address the following points.
1) There are many typos that need to be corrected.
a) Introduction, line 7: "...are a promising step..." (remove "the")
b) Line 86 (after Eq. 1): comma after the definition of V(r), kb should be k_{bond}, equality sign before 2.245{\sigma}_L
c) Lines 120 and 121: The radius of gyration should be R_g, not Rg2
d) Line 186: \Sigma_L=\piR_g^2/D^2
2) In Section 2, SI units for the various parameters of the simulation should be given (for pressure, temperature, density, and so on).
3) In the discussion of the grafting density (Section 3.4, line 183), we read that "we do not have uniformly distributed tethered chains." On the other hand, in Sec. 2.1 and Fig. 4, we see that the chains are tethered on a regular square lattice. This is an inconsistency that needs to be resolved.
4) I cannot understand the use of R_g instead of R_{gxy} in the definition of the local grafting density. I also don't find the local grafting density (and Fig. 8) particularly informative. It is just a rescaling of the radius of gyration. So, I believe it should be called rescaled radius of gyration in order to avoid confusion.
5) The definition of the island grafting density, \Sigma_{island}, is not clear at all. There are some statements in words, but a couple of formulas would be much clearer.
6) Since this article is submitted as a contribution to the Special Issue on "Semiflexible Polymers II," some discussion of semiflexibility would be nice. The authors are right to state that the elongated chains close to the center of the island behave in a semiflexible way, but a brief discussion of what semiflexibility is and how a finite bending stiffness would affect their results would enrich the paper.
Author Response
Thank you for your careful reading of the manuscript, we have addressed all your comments below, and hope that the manuscript reads better and is acceptable for publication..
In this article, the authors present a Molecular Dynamics simulation study of polymer islands (CORALs) of various sizes. A CORAL is an array of densely grafted polymers on a planar substrate with lateral size comparable to the length of a polymer chain. Their finite size makes them distinct entities from the polymer brushes. CORALs can have interesting applications, and their study is an important subject of research . This article is a significant contribution in the description of their conformational properties. I will recommend publication after the authors address the following points.
1) There are many typos that need to be corrected.
Thank you for catching these, we have carefully reread the entire manuscript and corrected the typos you graciously have pointed out as well as a few others.
2) In Section 2, SI units for the various parameters of the simulation should be given (for pressure, temperature, density, and so on).
These are scaled coordinates, so SI units of various parameters do not make sense, specifically if modeling polystyrene or polyethylene these parameters will scale to different SI units. This is done intentionally so that the physics of these grafted systems are captured in general, not any specific polymer.
3) In the discussion of the grafting density (Section 3.4, line 183), we read that "we do not have uniformly distributed tethered chains." On the other hand, in Sec. 2.1 and Fig. 4, we see that the chains are tethered on a regular square lattice. This is an inconsistency that needs to be resolved.
We have included text to clarify what we mean. Yes, the local density is homogeneous (e.g. we use squared pattern) but the reduced tethered density is NOT homogeneous (e.g. shown in Figure 9), and this is important to use for showing different regions will have different characteristics in terms of morphologies.
4) I cannot understand the use of R_g instead of R_{gxy} in the definition of the local grafting density. I also don't find the local grafting density (and Fig. 8) particularly informative. It is just a rescaling of the radius of gyration. So, I believe it should be called rescaled radius of gyration in order to avoid confusion.
In Figure 8, we use rescaled R_g to calculate the reduced tethered density. Then it is used to compare with previous polymer brush papers and show that the polymers in the island vary with position/distance from the edge. But it is overall greater than 5, which is brush-like behavior. However, the local reduced tethered density show difference in a different region (e.g. center, edge, outside)
5) The definition of the island grafting density, \Sigma_{island}, is not clear at all. There are some statements in words, but a couple of formulas would be much clearer.
We add definitions and formulas to clarify these issues.
6) Since this article is submitted as a contribution to the Special Issue on "Semiflexible Polymers II," some discussion of semiflexibility would be nice. The authors are right to state that the elongated chains close to the center of the island behave in a semiflexible way, but a brief discussion of what semiflexibility is and how a finite bending stiffness would affect their results would enrich the paper.
Agreed, we have added a paragraph/section to the discussion about semiflexible polymers and specifically our simulations in this context.
Reviewer 2 Report
Dear authors,
Your article reads well and very positively I noticed that you made an effort with the model validation and let the reader participate in it. I enjoyed reading your publication from 2018 (Reference 1), but would have liked to see a more intensive comparison of the simulation with the AFM measurements even there. In the publication now available three years later, if I have not missed anything, you have completely omitted a comparison with experimental findings. I therefore ask you to make up for this. Only in this way the quality of your simulations can be assessed comprehensibly for the reader. I therefore recommend the present manuscript for a major revision. Furthermore, it would be nice if you would give the community access to your simulation code.
Author Response
Your article reads well and very positively I noticed that you made an effort with the model validation and let the reader participate in it. I enjoyed reading your publication from 2018 (Reference 1), but would have liked to see a more intensive comparison of the simulation with the AFM measurements even there. In the publication now available three years later, if I have not missed anything, you have completely omitted a comparison with experimental findings. I therefore ask you to make up for this. Only in this way the quality of your simulations can be assessed comprehensibly for the reader. I therefore recommend the present manuscript for a major revision. Furthermore, it would be nice if you would give the community access to your simulation code.
Thank you for your thoughtful comments. However, the purpose of this manuscript is to characterize the semiflexible polymers with these systems, specifically within the islands. While we could add comparisons with experimental results such as AFM or conductivity measurements, this would be beyond the scope of this paper and the special issue for which it was submitted. We are preparing another manuscript for submission which will have more simulation and experimental data and comparison.
The code used is freely accessible to the community (we use lammps), we have provided all the necessary interaction for anybody to reproduce our work as they wish.
Reviewer 3 Report
Comments:
The manuscript entitled “Densely Packed Tethered Polymer Nanoislands: A Simulation Study“ presents structural properties of the coordinated responsive arrays of surface-linked polymer islands (CORALs) studied by the coarse-grained molecular dynamics. The variable parameter is the number of chains within an island ranging from one chain to infinite chains, i.e., to a continuous polymer brush. The chains are built up of 40 effective units with one effective unit corresponding to the persistence length. The solvent parameters are chosen to represent theta conditions for the grafted polymer chains. The different behavior of chains is found in the center, at the edge, and in the peripheral region of an island. The central part of an island resembles the classic polymer brush, while in the peripheral region, the chains exhibit features salient for an isolated tethered chain. The structural behavior of the polymers at the edge represent an intermediate between the polymer brush and isolated tethered chain. These conclusions are expectable and the informative value of the manuscript would considerable increase if the authors provide more findings for instance through comparison with the behavior of a CORALs system in a good and a poor solvent in one manuscript instead of splitting the study in separate manuscripts.
This manuscript may be considered for publication in Polymers journal if the and English are improved and also if the formal and factual mistakes and misprints are corrected.
The simulation model is not mesoscopic but coarse-grained and this needs to be corrected in the text.
How did the authors justify that the number of simulation steps was sufficient to achieve the system equilibration and reliable statistics of data during the production phase?
The paragraph (lines 68-73) is not clearly written. For instance, the authors write that they left the z-coordinate fluctuate perpendicularly to the tangent plane of the surface. Perpendicularly to the tangent plane means parallel to the plane. However, it follows from the data in the text that the z-coordinate is perpendicular to the plane. The following sentence is also misleading since the authors write that each individual cell dimension fluctuated around a constant value. These two sentences are in conflict. Moreover, if they let only the z-coordinate fluctuate they should use the semiisotropic control of pressure.
The points should be presented in Figure 3 because it is not clear whether there was a sufficiently large number of the data to draw reliable conclusions. The authors should explain in the text what the average, fit and derivative mean in the legend.
In line 79, the scaling of a free polymer chain in a good solvent should be close to N5/3. What is ro in the last expression of this line?
The readability of the whole paragraph (lines 97-104) should be improved.
The sentence in lines 108-109 is not clear.
In Table 2, it would be instructive to include the radius of the single chain as well.
In line 121, it should be stressed that the authors refer to the square of the radius of gyration instead of the radius of gyration and 2 should be presented as the exponent.
In section 3.1, the dimension of the grid elements should be given.
In Figures 6 and 7, there are no numbers associated with the x-coordinates.
The sentence in lines 136-137 is not clear.
The attempt to fit the last graph in Figure 7 with a parabola is not appropriate.
In line 146, B/w term should be defined.
In lines 150-152, the authors write that significant deviations from the parabolic profiles are found in smaller islands but opposite seems to be true in Figure 6.
How was the density scaled to provide the reduced density?
The data of the radius of gyration and the ratio of its parallel and perpendicular components should be presented in a table or in a graph.
The readability of section 3.4 should be improved. The parameter πσRg2 is defined to characterize the degree of crowding of tethered polymer chains and the radius of gyration of an isolated tethered chain or a free chain is assumed. Thus, the radius of gyration is supposed to be isotropic. It seems that the radius of gyration of the tethered chains is used in this section. If so, the square lateral component of the radius of gyration should be assumed.”
In Figures 8 and 9, the units are given in Å. There is no relation defined between the reduced units and Å units in the text. It should be defined.
In line 205, using the expression “collapsed state is not appropriate since it evokes the conformation of chains in a poor solvent.
The authors use the expression cites throughout the manuscript. Do they mean sites?
The brush regime is identified with “semiflexible”. The chain flexibility is the inherent chain property and a chain can behave as semiflexible even in the mushroom conformation.

Author Response
The manuscript entitled “Densely Packed Tethered Polymer Nanoislands: A Simulation Study“ presents structural properties of the coordinated responsive arrays of surface-linked polymer islands (CORALs) studied by the coarse-grained molecular dynamics. The variable parameter is the number of chains within an island ranging from one chain to infinite chains, i.e., to a continuous polymer brush. The chains are built up of 40 effective units with one effective unit corresponding to the persistence length. The solvent parameters are chosen to represent theta conditions for the grafted polymer chains. The different behavior of chains is found in the center, at the edge, and in the peripheral region of an island. The central part of an island resembles the classic polymer brush, while in the peripheral region, the chains exhibit features salient for an isolated tethered chain. The structural behavior of the polymers at the edge represent an intermediate between the polymer brush and isolated tethered chain. These conclusions are expectable and the informative value of the manuscript would considerable increase if the authors provide more findings for instance through comparison with the behavior of a CORALs system in a good and a poor solvent in one manuscript instead of splitting the study in separate manuscripts.
This manuscript may be considered for publication in Polymers journal if the and English are improved and also if the formal and factual mistakes and misprints are corrected.
Thank you for your careful reading of the manuscript, we have addressed all your comments below, and hope that the manuscript reads better and is acceptable for publication.
The simulation model is not mesoscopic but coarse-grained and this needs to be corrected in the text.
Coarse-grained is associated with combining atoms into a large site, where mesoscopic is typically associated with averaged interactions between molecules in a material. For example, there is a difference between a coarse-grain model of two different surfactants or polymers, where mesoscale there would be no difference. As our simulations fall into the later category we refer to them as mesoscopic in nature.
How did the authors justify that the number of simulation steps was sufficient to achieve the system equilibration and reliable statistics of data during the production phase?
We have added text to clarify our equilibrium process.
The paragraph (lines 68-73) is not clearly written. For instance, the authors write that they left the z-coordinate fluctuate perpendicularly to the tangent plane of the surface. Perpendicularly to the tangent plane means parallel to the plane. However, it follows from the data in the text that the z-coordinate is perpendicular to the plane. The following sentence is also misleading since the authors write that each individual cell dimension fluctuated around a constant value. These two sentences are in conflict. Moreover, if they let only the z-coordinate fluctuate they should use the semiisotropic control of pressure.
Fixed
The points should be presented in Figure 3 because it is not clear whether there was a sufficiently large number of the data to draw reliable conclusions. The authors should explain in the text what the average, fit and derivative mean in the legend.
Fixed
In line 79, the scaling of a free polymer chain in a good solvent should be close to N5/3. What is ro in the last expression of this line?
Fixed
The readability of the whole paragraph (lines 97-104) should be improved.
We are re-worded the paragraph to make it more readable.
The sentence in lines 108-109 is not clear.
In Table 2, it would be instructive to include the radius of the single chain as well.
The signal chain’s grafting is a single point that has no radius.
In line 121, it should be stressed that the authors refer to the square of the radius of gyration instead of the radius of gyration and 2 should be presented as the exponent.
Fixed
In section 3.1, the dimension of the grid elements should be given.
All units are reduced units of $\sigma_{LJ}$
In Figures 6 and 7, there are no numbers associated with the x-coordinates.
Thanks for pointing this out, they have been fixed.
The sentence in lines 136-137 is not clear.
This has been reworded.
The attempt to fit the last graph in Figure 7 with a parabola is not appropriate.
We agree that it is not appropriate, and we wanted to point that out.
In line 146, B/w term should be defined.
Fixed.
In lines 150-152, the authors write that significant deviations from the parabolic profiles are found in smaller islands but the opposite seems to be true in Figure 6.
Thanks for catching this, we had left out a negative.
How was the density scaled to provide the reduced density?
We have put in the formula we used to get the reduced density.
The data of the radius of gyration and the ratio of its parallel and perpendicular components should be presented in a table or in a graph.
Figure 8 is a presentation of the parallel and perpendicular components. We have tried to clarify this in the text. The individual components do not add anything to the paper, and the reduced Rg{z/XY} is more universal, which we have reported.
The readability of section 3.4 should be improved. The parameter πσRg2 is defined to characterize the degree of crowding of tethered polymer chains and the radius of gyration of an isolated tethered chain or a free chain is assumed. Thus, the radius of gyration is supposed to be isotropic. It seems that the radius of gyration of the tethered chains is used in this section. If so, the square lateral component of the radius of gyration should be assumed.”
We have modified section 3.4 to be more readable.
In Figures 8 and 9, the units are given in Å. There is no relation defined between the reduced units and Å units in the text. It should be defined.
Fixed
In line 205, using the expression “collapsed state is not appropriate since it evokes the conformation of chains in a poor solvent.
This has been removed.
The authors use the expression cites throughout the manuscript. Do they mean sites?
Fixed (got caught in a global replace).
The brush regime is identified with “semiflexible”. The chain flexibility is the inherent chain property and a chain can behave as semiflexible even in the mushroom conformation.
True. Flexibility increases as you move from the center towards the edge of the islands. The center is semiflexible (brush like) and the edge is fully flexible. Mushroom confirmation would be between these two states - aka less flexible than the brush but more flexible than the brush.
Round 2
Reviewer 1 Report
I find the revised manuscript significantly improved with respect to the original version. I will recommend publication after the authors fix one last issue. I agree that the use of scaled units instead of SI units may be more appropriate. However, a clear and explicit definition of these scaled units must be included.
Author Response
Thank you again for your helpful comments. We have updated the manuscript to included explicit scaling for some polymers. Hopefully this will clarify the scaling and be helpful to the reader.
Reviewer 2 Report
Thank you very much for the revision. You have significantly increased the value for the interested reader.
Author Response
Thanks for you helpful comments on the first version.